# Spatiotemporal Variation in Water Deficit- and Heatwave-Driven Flash Droughts in Songnen Plain and Its Ecological Impact

Jiahao Sun [1,2,3,†], Yanfeng Wu [3,†], Qingsong Zhang [1,2], Lili Jiang [3], Qiusheng Ma [1,2,3], Mo Chen [1,2,*], Changlei Dai [1,2] and Guangxin Zhang [3]

1   Institute of Groundwater in Gold Region, Heilongjiang University, Harbin 150080, China; hydro110@163.com (J.S.); zhangqingsong@iga.ac.cn (Q.Z.); maqiusheng@iga.ac.cn (Q.M.); 2006003@hlju.edu.cn (C.D.)
2   School of Hydraulic and Electric-Power, Heilongjiang University, Harbin 150080, China
3   State Key Laboratory of Black Soils Conservation and Utilization, Northeast Institute of Geography and Agroecology, Chinese Academy of Sciences, Changchun 130102, China; wuyanfeng@iga.ac.cn (Y.W.); jianglili@iga.ac.cn (L.J.); zhgx@iga.ac.cn (G.Z.)
*   Correspondence: chenmocc12@163.com
†   These authors contributed equally to this work.

**Abstract:** The phenomenon of flash droughts, marked by their fast onset, limited predictability, and formidable capacity for devastation, has elicited escalating concern. Despite this growing interest, a comprehensive investigation of the spatiotemporal dynamics of flash drought events within zones of ecological sensitivity, alongside their consequential ecological ramifications, remains elusive. The Songnen Plain, distinguished as both an important granary for commodity crops and an ecological keystone within China, emerges as an indispensable locus for the inquiry into the dynamics of flash droughts and their repercussions on terrestrial biomes. Through the application of daily soil moisture raster datasets encompassing the years 2002 to 2022, this investigation delves into the spatiotemporal progression of two distinct categories of flash droughts—those instigated by heatwaves and those precipitated by water deficits—within the Songnen Plain. Moreover, the ecosystem's response, with a particular focus on gross primary productivity (GPP), to these climatic variables was investigated. Flash drought phenomena have been observed to manifest with a relative frequency of approximately one event every three years within the Songnen Plain, predominantly lasting for periods of 28–30 days. The incidence of both heatwave-induced and water deficit-induced flash droughts was found to be comparable, with a pronounced prevalence during the summer and autumn. Nevertheless, droughts caused by water scarcity demonstrated a more extensive distribution and a heightened frequency of occurrence, whereas those rooted in heatwaves were less frequent but exhibited a propensity for localization in specific sectors. The sensitivity of GPP to these meteorological anomalies was pronounced, with an average response rate surpassing 70%. This spatial distribution of the response rate revealed elevated values in the northwestern segment of the Songnen Plain and diminished values towards the southeastern sector. Intriguingly, GPP's reaction pace to the onset of heatwave-driven flash droughts was observed to be more rapid in comparison to that during periods of water scarcity. Additionally, the spatial distribution of water use efficiency during both the development and recovery periods of flash droughts largely deviated from that of base water use efficiency. The insights from this study hold profound implications for the advancement of regional drought surveillance and adaptive management.

**Keywords:** soil moisture; flash drought; spatiotemporal evolution; ecological impacts; Songnen Plain

## 1. Introduction

In the ambit of China's climatic tribulations, drought reigns as the most ubiquitous and pernicious natural calamity, distinguished by its insidious inception, protracted duration, and extensive influence [1–3]. Amidst the shifting paradigms of global climate change,

there has been a notable escalation in both the frequency and severity of droughts, yielding increasingly adverse repercussions upon societal, economic, and ecological spheres [4,5]. Of late, a novel classification of drought phenomena—the emergent 'flash droughts'—has garnered escalating scholarly intrigue. These flash drought episodes, in stark contrast to their traditional counterparts, are characterized by a fast onset, diminished forecast ability, and heightened disruptive potential [6,7]. Often manifesting in concert with sustained periods of deficient precipitation, heatwaves, robust winds, and intense solar radiation, these flash droughts precipitate a swift diminution in soil moisture levels, thereby inflicting profound detriments upon agriculture and natural ecosystems [8,9]. As the specter of global warming looms, these flash droughts may well evolve into the prevalent paradigm of global aridity [3,10,11]. Consequently, there is an exigent imperative to delve into the spatiotemporal dynamics and underlying drivers of these flash droughts [12], particularly within regions that are ecologically fragile and climatically sensitive.

Internationally, scholars have embarked on an extensive array of research endeavors investigating the spatial–temporal nuances, catalysts, and prospective trajectories of flash drought phenomena at global, national, and watershed scales, as well as across diverse biogeographical realms, culminating in a wealth of scholarly findings [2,13,14]. Predominantly, these academic inquiries have concentrated on the ramifications of flash droughts upon agriculture and ecological systems, elucidating the underlying mechanisms of their impact [3,15]. For instance, Mo and Lettenmaier delineated flash droughts based on fluctuations in temperature, evapotranspiration (ET), and soil moisture throughout drought evolution [16]. They bifurcated them into two distinct categories predicated on their causative mechanisms: one category encompasses heatwave-induced droughts, primarily instigated by elevated temperatures that augment ET and diminish soil moisture levels [13]; the other encompasses precipitation deficiency droughts, principally propelled by a paucity of rainfall, which similarly escalates ET and reduces soil moisture [17]. This latter category also includes droughts arising from a synergy of decreased rainfall-induced ET and anomalous thermal elevation. In dissecting these two manifestations of flash droughts, scholars have delved into their characteristics and driving forces through empirical temperature analyses and the modeling of soil moisture and ET reconstructions [18,19]. Wang et al. leveraged meteorological data from the China Meteorological Administration (CMA) alongside reanalysis products of soil moisture data to probe into the enduring shifts and causatives of high-temperature-induced flash droughts within China, discerning that the rising trend of such droughts is predominantly linked to long-term climatic warming [20]. Yuan et al. ascertained that, over the preceding six decades, the incidence of droughts driven by rainfall scarcity has seen a threefold increase in southern Africa, particularly during the 2015 and 2016 rainy seasons, precipitating a diminution in ecosystem carbon sequestration exceeding 100 TgC [21]. The influence of droughts on terrestrial ecosystems has increasingly come under scrutiny in recent times. Yuan et al. ascertained that, within the final quarter-century (2075–2099), the detrimental impacts of drought on terrestrial ecosystems are poised to nearly triple under scenarios of both high and medium greenhouse gas (GHG) emissions [22]. Nevertheless, the precise manner and extent to which flash droughts impinge upon terrestrial ecosystems remain to be fully elucidated. Moreover, understanding the differential responses of various ecosystems to flash droughts can significantly bolster predictions regarding the evolution of ecosystems amidst global climate change, their spatial–temporal dynamics, and serve as a critical reference for formulating drought mitigation and prevention strategies.

Northeast China, recognized as a region acutely sensitive to climatic vicissitudes, emerges as a crucible for drought susceptibility and a focal point for global drought research [6,23]. The Songnen Plain (Figure 1), heralded for its quintessential black soil, stands as a pivotal agricultural nexus and a bastion of wetland biodiversity in Northeast China [24]. The trajectory of drought within this locale is inexorably linked to the imperatives of regional food security and ecological integrity. In the wake of global climate change, the area has witnessed a marked escalation in the frequency of drought episodes, exacerbating

the ecological vulnerability of this region [25,26]. Flash droughts, characterized by their swift onset and profound ecological ramifications, represent a formidable natural calamity. Yet, the specific ecological consequences of flash drought events on the Songnen Plain remain an enigmatic aspect of study. This research endeavor focuses on the Songnen Plain to dissect the spatial patterns of occurrence and duration of two distinct flash drought modalities (namely, water scarcity-induced flash droughts and heatwave-induced flash droughts), alongside their seasonal and interannual variations. Further, it delves into the repercussions of these flash droughts on gross primary productivity (GPP), a probe of critical significance for charting the course of sustainable development in the Songnen Plain, underpinned by the principles of water security.

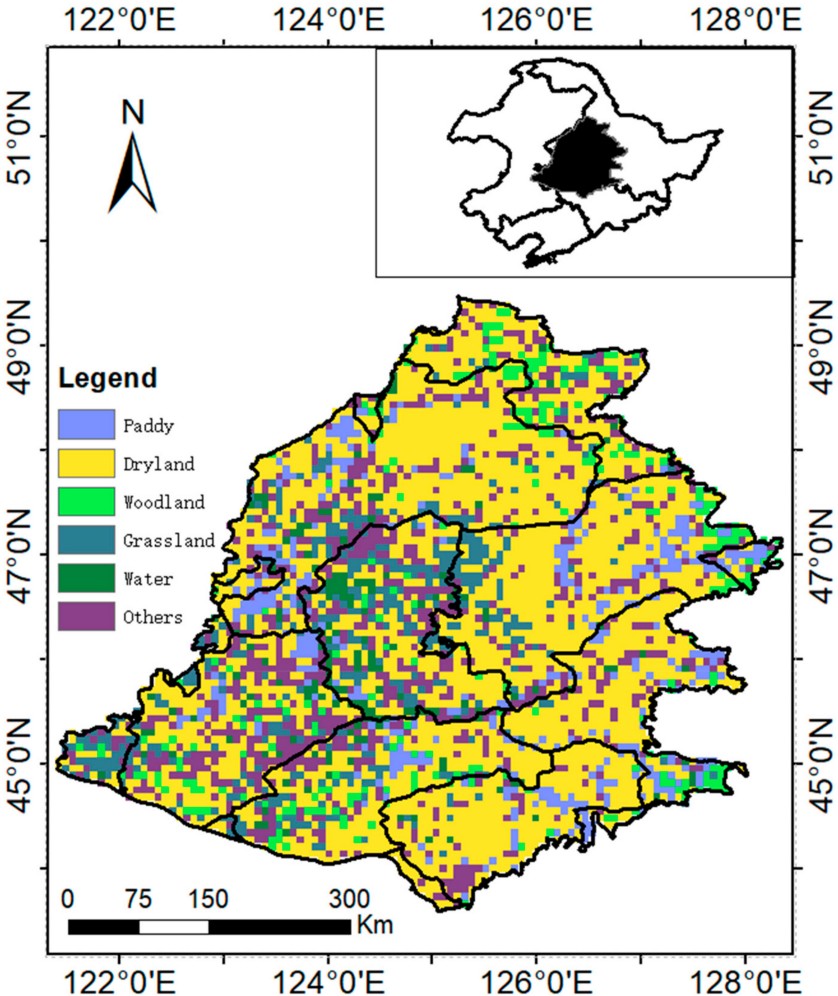

**Figure 1.** A map of the Songnen Plain.

## 2. Data and Methodology

### 2.1. Data

The NNsm (https://data.tpdc.ac.cn, accessed on 1 May 2023) soil moisture raster dataset released by the National Tibetan Plateau Science Data Center was used in this study. The data have a spatial resolution of 36 km × 36 km and a time span of 2002–2022 [27]. NNsm was able to accurately reproduce the SMAP (Soil Moisture Active Passive) surface soil moisture content with a global mean error of 0.029 $m^3/m^3$ [28]. Following Rigden et al. [29], the corresponding root zone moisture was estimated by NNsm using an autoregressive exponential framework combined with soil properties in the study area, which also compared well with in situ observations of surface soil moisture content and outperformed JAXA's (Japan Aerospace Exploration Agency's) and LPRM's (Land Param-

eter Retrieval Model's) AMSR-E/2 standard soil moisture data products [30]. Potential evapotranspiration data were obtained from the Global Land Evapotranspiration Model Amsterdam (GLEAM v3) (https://www.gleam.eu/, accessed on 1 May 2023). The Priestley and Taylor equation in GLEAM v3 calculates potential evaporation based on observations of surface net radiation and near-surface air temperature. GLEAM was used to maximize the recovery of evapotranspiration-related information from existing data stacks derived from space-based climate and environmental observations [29]. Vapor pressure deficit (VPD) data were obtained from the Community Long-Term Infrared Microwave Coupled Atmospheric Product System version 2 Level 3 product. The GPP dataset was derived from the Zheng et al. [30] release of the 0.05° spatially resolved and 8-day temporally resolved 1982–2018 global GPP dataset. All the above data were resampled using a bilinear interpolation method and uniformly processed to a resolution of 36 km × 36 km.

### 2.2. Methodology

#### 2.2.1. Ascertainment of Quintile Rankings for Pentad Soil Moisture Levels

The heterogeneity of soil moisture, underscored by pronounced regional disparities and seasonal fluctuations, necessitates a nuanced approach to encapsulate the transient states of soil hydration [31,32]. Traditional metrics, such as soil water depth or volumetric moisture content, offer snapshots of soil moisture yet falter in their utility for longitudinal drought analyses over expansive temporal and spatial domains [7,10]. To surmount this limitation, the present study advocates for the transformation of soil moisture readings into percentile rankings [2], thereby augmenting the spatial–temporal coherence in drought identification outcomes. Herein, we introduce a dual-faceted percentile framework, comprising the following: (a) the annual pentad average soil moisture percentile, recalibrated annually, and (b) the comprehensive time series pentad average soil moisture percentile. The former aims to mirror the moisture conditions across identical temporal windows within a given region, enhancing intra-annual comparability, whereas the latter seeks to illuminate shifts in soil moisture across successive intervals within the same locale, bolstering inter-temporal analysis.

The methodological blueprint for calculating the annual pentad average soil moisture percentile unfolds as follows: (1) implementing a pentad moving average on soil moisture data to smooth short-term variability; (2) segmenting the chronology into 365 distinct pentad series (leap year's February 29 excluded), thereby stratifying the dataset into quintiles on a pentad basis to distill soil moisture percentiles; and (3) synthesizing these 365 percentile sequences in chronological succession to fabricate a unified, extended time series. The computation of the comprehensive time series pentad average soil moisture percentile adheres to a similar procedural paradigm, albeit anchored to the entirety of the time series, ensuring a consistent methodology across both percentile derivations.

#### 2.2.2. Elucidation of Flash Drought Phenomena and Their Commencement Phases

This investigation delineates flash drought occurrences through a dual-lensed perspective, focusing on the temporal evolution of soil moisture content alongside the duration of aridity episodes. Drawing upon precedents set by Yuan et al. [2], this study synthesizes criteria encompassing both the precipitous decline in soil hydration levels and the sustained period of drought to characterize flash drought events. A soil moisture percentile below 40% signals the initial phase of departure from normative hydration states, serving as the preliminary demarcation of the flash drought sequence; when the soil moisture percentile descends beneath the 20% mark, it signifies substantial ecological distress. This latter threshold, endorsed by the U.S. Drought Monitor as indicative of "Moderate Drought—D1" conditions, underscores a critical juncture at which vegetative systems may exhibit signs of compromised water transport mechanisms [33].

Hence, the 20% threshold is adopted as the critical lower limit for identifying the progression into a drought condition. Leveraging the methodology of Yuan et al. [2], our analysis not only captures the onset and cessation of flash droughts but also emphasizes

their swift escalation and duration as definitional hallmarks. Specifically, a flash drought is recognized by a swift reduction in the average soil moisture percentile from above 40% to below 20%, at a decline rate exceeding 5%, persisting for a minimum of 15 days (Figure 2). The termination of a flash drought is marked by the recovery of soil moisture percentile levels above 20% [10]. The criteria for determining such events are as follows: (1) for an event to qualify as a flash drought, the soil moisture percentile must not exceed 40% at any point, with at least 15 consecutive days where the soil moisture remains at or below the 20% threshold; (2) the temporal span from the onset to the conclusion of a flash drought is categorized based on calendar duration, with the inception defined by the last instance above the 40% percentile and cessation as the final occurrence below the 20% threshold; and (3) the initiation phase of a flash drought necessitates a rapid fall to the 20% percentile within 20 days or fewer, characterized by an average decrement rate surpassing 5%.

$$Intensification\ rate: \begin{cases} \dfrac{\mathrm{SM}(t_{i+1}) - \mathrm{SM}(t_i)}{t_{i+1} - t_i} \geq 5\ \text{th percentile} \\ 0 < t_e - t_o \leq 5 \end{cases} \tag{1}$$

$$Drought\ condition: \begin{cases} \mathrm{SM}(t_o) \geq 40\ \text{th percentile} \\ \mathrm{SM}(t_e) \leq 20\ \text{th percentile} \\ t_p - t_e \geq 3 \end{cases} \tag{2}$$

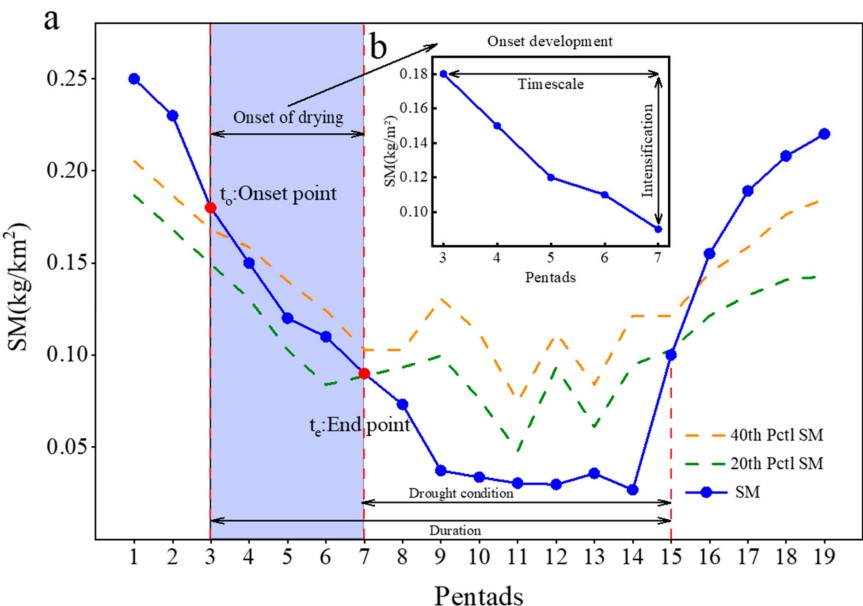

**Figure 2.** A schematic representation of the method used to identify a flash drought event. The subfigure shows the development period of flash drought revealed by changes in soil moisture.

Unlike the broader process of drought identification, the incipience of a flash drought is demarcated by the initial 20 instances wherein the soil moisture percentile first descends to the 20% threshold, marking the preliminary phase of the drought's duration. Consequently, recognizing the onset of a flash drought acts as a harbinger for discerning its persistence phase. To circumvent scenarios of extreme percentile fluctuations or the sustained presence of drought conditions within the 30–40% percentile range, the criterion for initiating the flash drought identification employs the last occurrence among three successive instances where soil moisture percentiles exceed 40%, within the first 22 days of the persistence phase.

### 2.2.3. Criteria for Differentiating Water Scarcity- and Heatwave-Induced Droughts

Given the heterogeneity of vegetation coverage and the distinct topographical features of the Songnen Plain [34], it became imperative to undertake separate spatial and tempo-

ral examinations of heatwave-induced and water scarcity-induced flash droughts. The mechanism triggered by a flash drought can be classified into two categories [13,17]: One is precipitated by short-term high-temperature weather, specifically heatwaves that intensify soil evaporation and precipitate a sharp decrease in soil water content. The second type of flash droughts is attributed to a regional precipitation deficiency, resulting in an insufficient soil water content, and weakened soil evapotranspiration capacity, coupled with abnormally high air temperatures. More precisely, when the air temperature is anomalously high, accompanied by low soil moisture content and evapotranspiration exceeding the norm, it is classified as a heatwave-induced flash drought. Conversely, when the air temperature remains abnormally high but the soil moisture content is low and precipitation falls below the standard range, it is identified as a water scarcity-induced flash drought [19,35].

The type of flash droughts was identified according to the following procedure:

(i) Temperature condition judgment: Determine whether the temperature anomaly of a certain meteorological day is higher than 1.0 times the standard deviation.

(ii) Determination of the ET conditions: If the air temperature is abnormally high, the soil moisture is low, and the ET anomaly is positive, it is a heatwave-induced flash drought; otherwise, if the air temperature is abnormally high, the soil moisture is low, and the ET anomaly is negative, it is a water scarcity-induced flash drought.

### 2.2.4. Ecological Impacts of Flash Drought Episodes

GPP constitutes the foundational element for global vegetative growth and alimentary production, occupying a pivotal role in the modulation of atmospheric carbon dioxide levels through its influence on the carbon balance within ecosystems [15]. Drought emerges as the most prevalent agent impinging upon GPP, markedly affecting it by modulating ecosystem respiration and plant photosynthetic processes, thereby injecting considerable uncertainty into forecasts of future terrestrial carbon sequestration capacities [36,37]. GPP encapsulates the aggregate volume of photosynthetic carbon dioxide fixation at the ecosystem scale, impacting all variables within the carbon cycle [38–40]. The process of photosynthetic carbon dioxide assimilation is modulated by factors such as the leaf area index, Rubisco enzyme activity, and stomatal conductance [41]. From an ecological standpoint, a negative deviation in GPP signals the initiation of an ecological response, with standardized anomalies being computed as follows [42,43]:

$$GPPSA = (GPP - \mu_{Gpp})/\sigma_{GPP} \tag{3}$$

where *GPPSA* represents the standardized anomalies of *GPP*, with $\mu_{Gpp}$ and $\sigma_{GPP}$ denoting the standard deviation and mean of the GPP time series, respectively. This investigation utilizes response time indices and frequency metrics to probe the interplay between ecological responses and flash drought events. The response rates for each grid are ascertained by the ratio of flash drought occurrences with negative GPPSA to the aggregate count of flash droughts. A diminished response frequency suggests a reduced ecological risk, and conversely, a higher frequency indicates elevated vulnerability. The response time is delineated as the sequence of positive standardized anomalies during a flash drought until the emergence of the initial negative anomaly. The calculation equation for the response rates (Rate) is as follows:

$$Rate = \frac{n_{GPPSA_{negtive}}}{N} \tag{4}$$

where $n_{GPPSA_{negtive}}$ refers to the number of flash droughts with negative *GPPSA*; *N* shows the aggregate count of flash droughts.

The carbon and water cycles are intricately interconnected via stomatal pathways, with vegetation adopting mitigative strategies to counteract drought stress, such as enhancing water use efficiency (WUE, defined as GPP/ET) [44,45]. Beyond drought, WUE is also influenced by Vapor Pressure Deficit (VPD). Hence, this study employs the underlying water use efficiency (uWUE) concept introduced by Zhou et al. (2020) to examine the sensitivity of the ecosystem's WUE to VPD fluctuations [46]. uWUE, computed as the

ratio of (GPP $\times \sqrt{GPP}$) to ET, accounts for the effects of VPD and offers a more accurate depiction of plant biochemical functions compared to traditional WUE metrics. It is computed as the ratio of GPP to potential evapotranspiration (PED), capturing the nonlinear dynamics between PED, VPD, and GPP. Variations in uWUE are closely aligned with drought conditions, with more severe droughts eliciting pronounced, positive anomalies in uWUE metrics. Consequently, deviations in uWUE are positively correlated with the intensity of the drought, underscoring the magnitude of potential positive anomalies in uWUE in response to severe drought conditions [15,43,47].

This study calculates annual averages of WUE and uWUE based on daily values, with standardized anomalies for WUE and uWUE determined in a manner akin to the equation, facilitating the assessment of anomaly impacts. This research examined the response of GPP to flash drought events spanning from 2002 to 2018. Given the disparity in time resolution between GPP and flash droughts, 8-day GPP was linearly transformed into 5-day GPP using linear interpolation for a more precise correspondence analysis of GPP's response to flash droughts.

## 3. Results

### 3.1. Subsection

Spatiotemporal Distributions of Flash Drought Events

The spatiotemporal patterns of flash drought occurrences across the Songnen Plain, as delineated in Figure 3, reveal that the apex of flash drought episodes between 2002 and 2022 reached a tally of 13, with the protracted duration extending up to 34 days. The expanse affected by these flash arid conditions is considerably vast, albeit with a heterogenous spatial distribution. Predominantly, the urban locales of Baicheng and Songyuan in western Jilin Province, along with Suihua in Heilongjiang Province, are more susceptible, experiencing such events approximately every three years, with occurrences exceeding seven times. Most of these flash droughts spanned durations of 28–30 days, devoid of regions enduring longer spells.

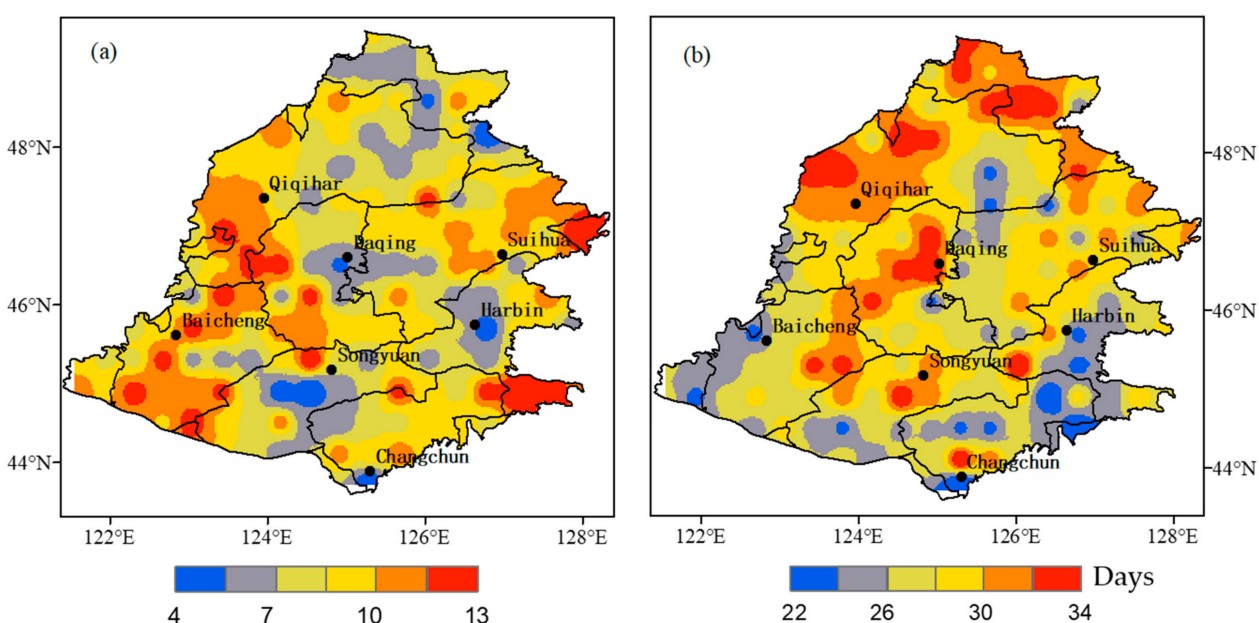

**Figure 3.** Spatial distribution of the (**a**) number and (**b**) average duration of flash drought events in the Songnen Plain during 2002–2022.

### 3.2. Spatial Distribution of Water Deficit- and Heatwave-Driven Flash Droughts

To further dissect the evolution of flash drought phenomena within the Songnen Plain, this analysis bifurcates the events into water scarcity- and heatwave-driven events,

delving into the spatiotemporal evolution of these distinct drought typologies. Figure 4 illustrates the spatial variances in the frequency and mean duration of both drought categories from 2002 to 2022, unveiling pronounced disparities. The water scarcity droughts exhibit a high frequency and widespread distribution, encapsulating a spatial motif of peripheral prevalence yet central scarcity. Notably, over 80% of the northeastern regions near Suihua and Qiqihar were afflicted by water scarcity droughts. Conversely, heatwave-driven droughts were more localized, with certain areas registering 8–10 events. Regarding duration, water scarcity droughts averaged around 26 days, occasionally extending to 45 days, whereas heatwave-driven droughts averaged 28 days, diminishing to approximately 22 days in western Jilin. Thus, the Songnen Plain is a battleground for both water deficit- and heatwave-driven droughts, with the former being more frequent but the latter enduring longer.

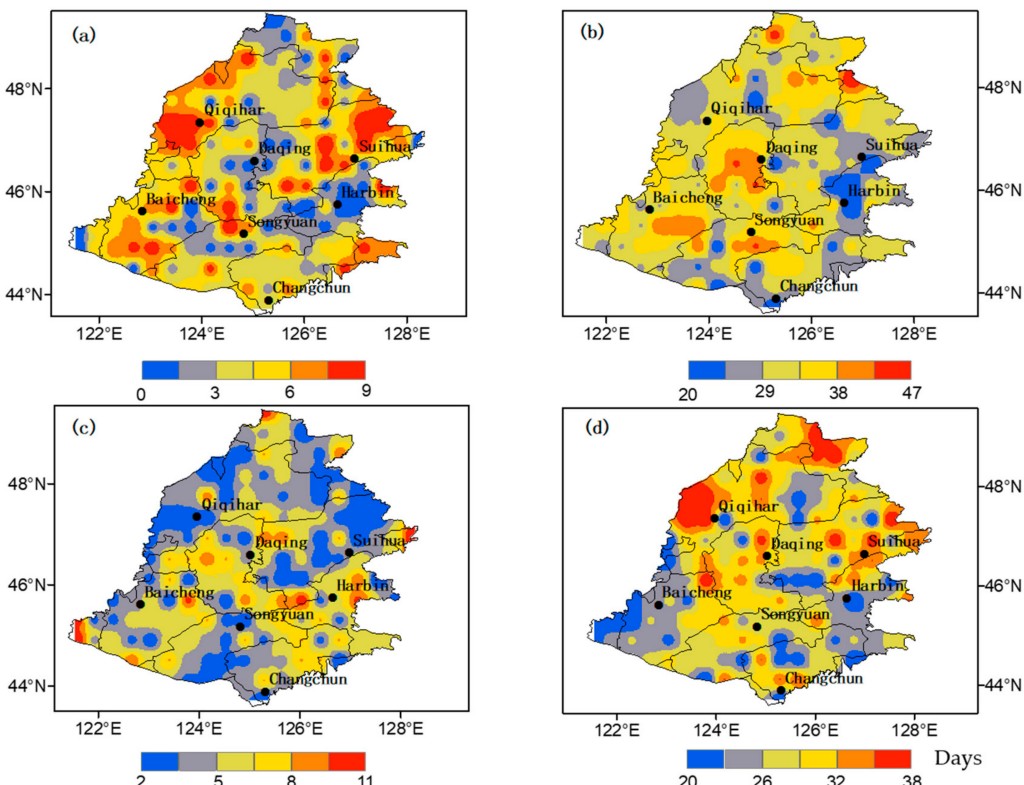

**Figure 4.** Spatial distribution of the number (**a**,**c**) and average duration (**b**,**d**) of water scarcity- (the first line) and heatwave-driven (the second line) flash droughts in the Songnen Plain during 2002–2022.

### 3.3. Temporal Changes in Water Deficit- and Heatwave-Driven Flash Droughts

A temporal analysis (Figure 5) indicates a general decline in both the frequency and duration of flash droughts from 2002 to 2022, with the longest span of water scarcity droughts (110 days) surpassing that of heatwave-driven droughts (85 days). Seasonally, both drought types predominantly occurred during summer and fall, with the most extended events, surpassing 50 days, concentrated in summer, followed by fall. The inter-annual frequency exhibited a fluctuating pattern of decline, rise, and then decline.

Seasonal variation analyses of both drought types from 2002 to 2022 (Figure 6) revealed their presence across spring, summer, and fall, with exceptions in spring 2002 and 2012. Summer witnessed the highest prevalence, with an average area ratio of 26% across multiple years and over 20% of the study area affected in 9 out of 20 years. Furthermore, the area ratios of seasonal flash droughts exhibited significant yearly disparities; for instance, in 2007, 73% of droughts transpired in summer, contrasted by 14% in spring and 21% in fall. Notably, 2002 and 2012 recorded no spring droughts, but witnessed occurrences in summer

and fall. The area ratios for spring, summer, and fall droughts showcased a trend of initial decline followed by an increase, with a downturn from 2002 to 2011 and an upturn from 2012 to 2022. Comparative scrutiny reveals that water scarcity droughts predominantly occur in spring, whereas heatwave-driven droughts are more frequent in summer and fall, with fall 2004 witnessing 42% of the area affected by water deficit droughts, juxtaposed against 21% by heatwaves.

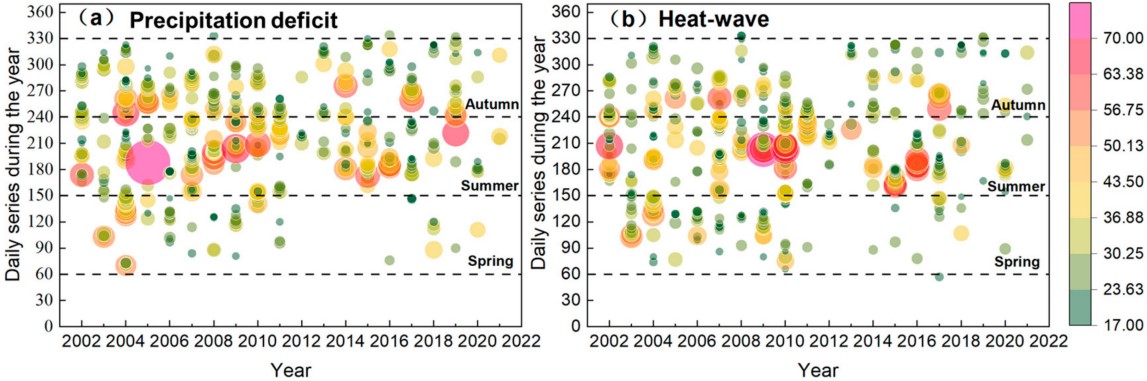

**Figure 5.** Bubble plots of the duration (days) and onset time of (**a**) water deficit- and (**b**) heatwave-driven flash drought events during 2002–2022.

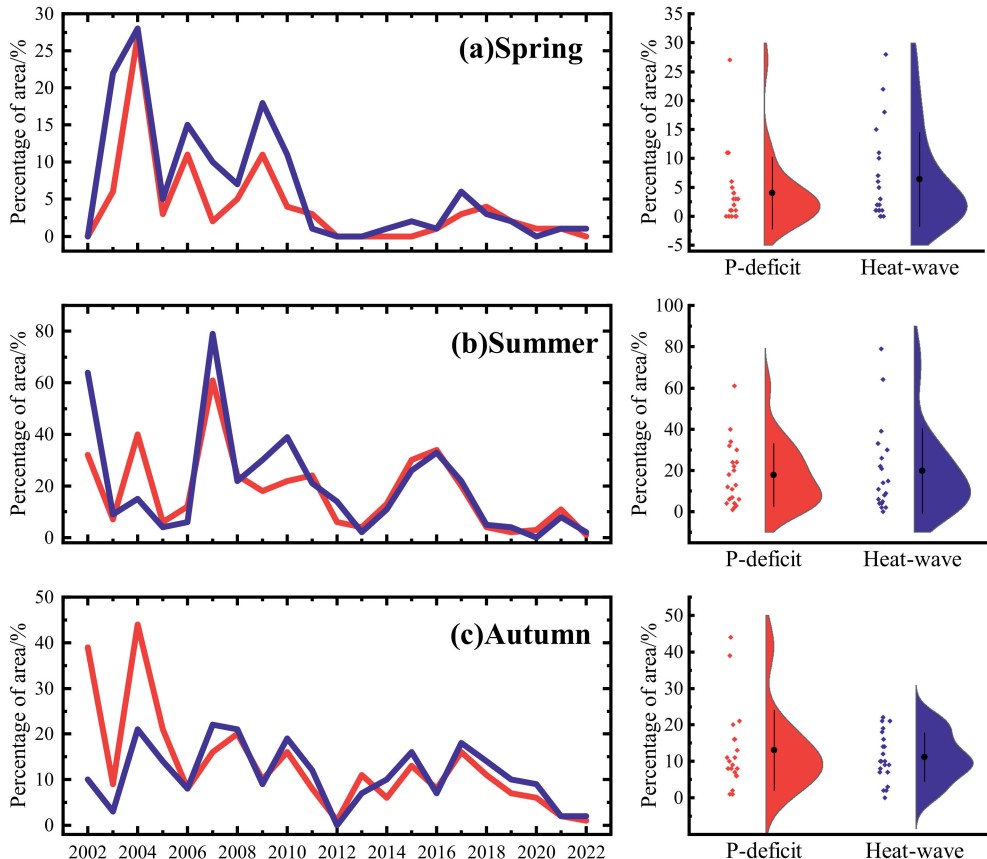

**Figure 6.** Annual variation (the left panel), data distribution, and probability density of areas' proportion (the right panel) affected by water deficit (P-deficit)- and high-temperature (Heat-wave)-driven flash drought events during the spring, summer, and autumn in the Songnen Plain from 2002 to 2022. The red and purple color refer to P-deficit and Heat-wave driven flash drought events respectively.

### 3.4. Ecological Response to Flash Droughts

Figure 7 elucidates the spatial distribution of the GPP response rates to two distinct categories of flash drought events across the Songnen Plain from 2002 to 2022. It can be observed that water deficit- and high-temperature-driven flash drought showed similar spatial dynamics, with a pronounced concentration in the northwest as opposed to the southeast. Within this framework, areas exhibiting a high response rate to water scarcity droughts are extensively distributed, showcasing an average response rate of 76%, and nearly 47% of this terrain is characterized by a response rate exceeding 80%. Conversely, regions affected by heatwave-driven droughts are more localized, maintaining an average response rate of 70%, with 37% of these areas exhibiting response rates above 80%.

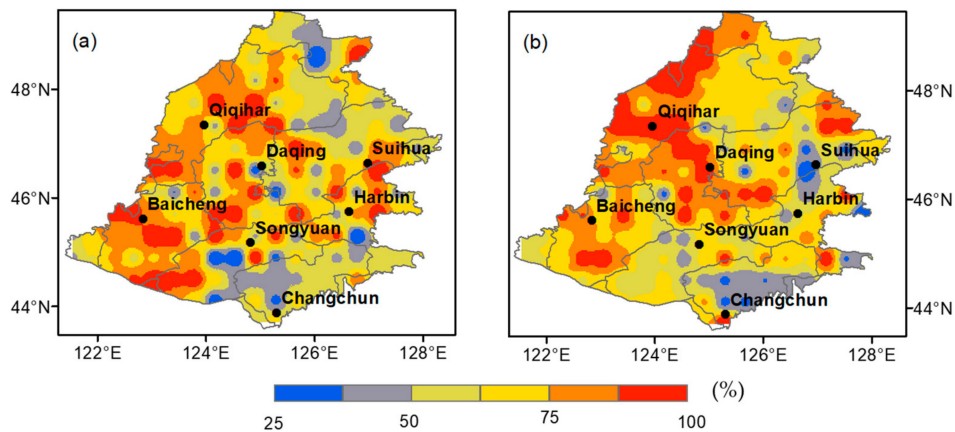

**Figure 7.** The spatial distribution of the GPP response rates to (**a**) water deficit- and (**b**) heatwave-driven flash drought events across the Songnen Plain from 2002 to 2022.

In comparison to GPP, WUE provides a more refined reflection of the ecosystem's water–carbon interplay in reaction to flash drought phenomena. Figure 8 delineates the normalized anomalies of WUE and its underlying variant (uWUE) amidst the onset and amelioration of flash drought conditions, illustrating that positive standardized anomalies of uWUE surpass those of WUE during both the escalation and recuperation phases. Notably, both WUE and uWUE register significant negative anomalies throughout the escalation of both drought types. Spatially, the negative standardized anomalies of WUE associated with water scarcity are predominantly concentrated in the northern reaches of the Songnen Plain and notably within Qiqihar and Daqing cities during the escalation phase (Figure 8a), extending into the northern and central zones during the recuperation phase (Figure 8b). In contrast, heatwave-driven anomalies are centrally and westerly focused during the escalation phase, showcasing a more confined distribution but with heightened positive anomalies (Figure 8c), and during the recuperation phase, negative anomalies are primarily localized in the central and western regions (Figure 8d).

When juxtaposing uWUE's anomalies against those of WUE, it becomes evident that water scarcity flash droughts exhibit superior overall performance during the escalation phase (Figure 8e), with the recuperation phase witnessing more pronounced and geographically concentrated positive anomalies within the western sector of the Songnen Plain (Figure 8f). The spatial distribution of uWUE anomalies during the heatwave-driven escalation phase mirrors that of WUE (Figure 8g), albeit with an expanded distribution of positive anomalies during the recuperation phase (Figure 8h). Moreover, an examination reveals that positive anomalies of WUE and uWUE are more prevalent during heatwave-driven droughts compared to water scarcity ones, indicating a swifter GPP response to the former drought type. Throughout the drought recuperation phase, certain locales, including the Qiqihar and Songyuan areas, persist with negative anomalies in WUE and uWUE, signaling a diminished ecological resilience within the vegetation of these regions.

These findings suggest a gradual erosion of the ecological self-regulatory capacity in these areas, exacerbated by the protracted duration of drought events.

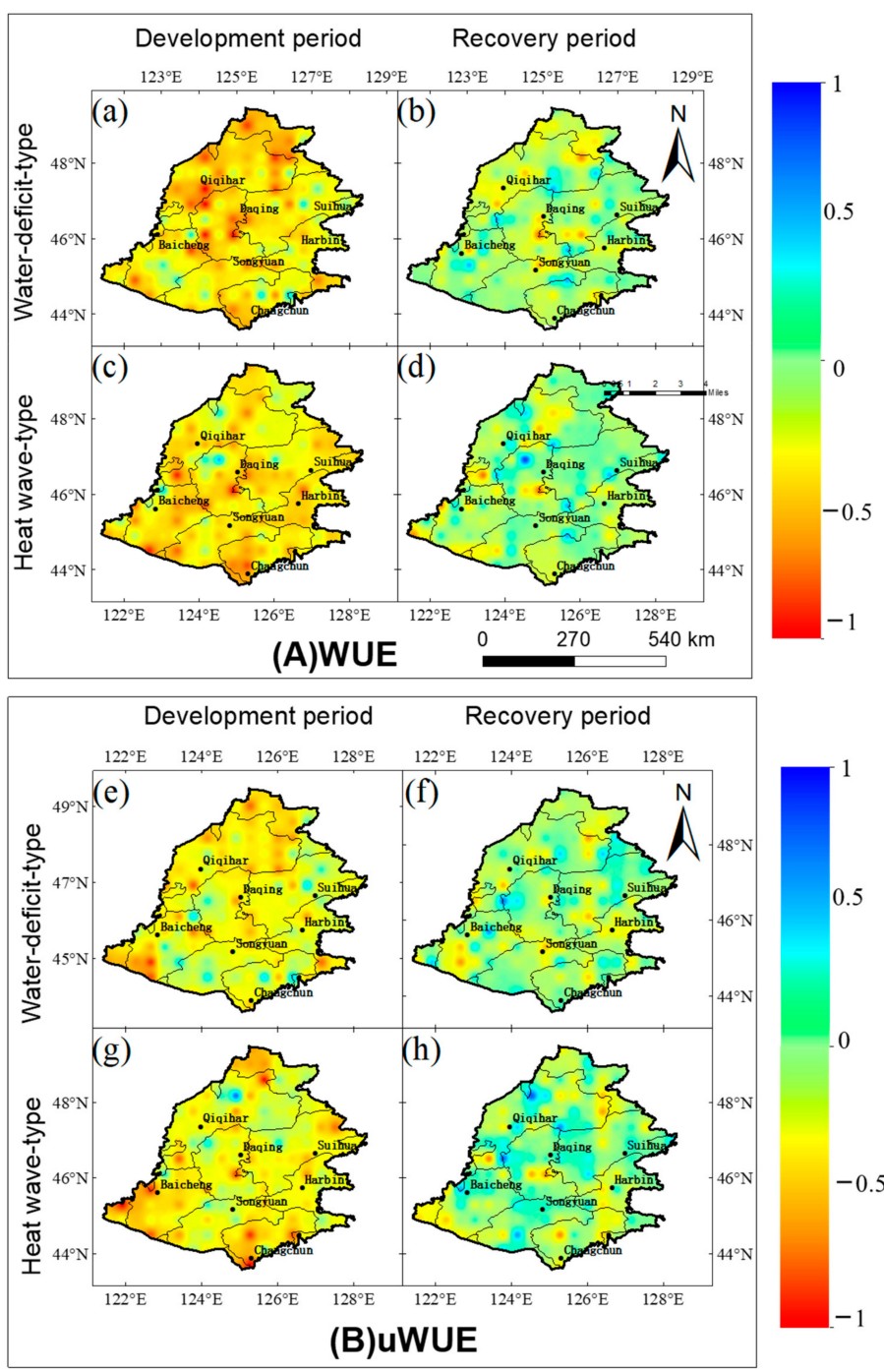

**Figure 8.** Spatial distribution of water use efficiency (WUE) versus base water use efficiency (uWUE) during the development (**a**,**c**,**e**,**g**) and recovery periods (**b**,**d**,**f**,**h**) of water deficit-type and heatwave-driven flash droughts events from 2002 to 2022.

## 4. Discussion

Amidst escalating concerns and formidable challenges posed by flash droughts, the evolution and consequential impact of these phenomena on the terrestrial ecosystems of the Songnen Plain—a pivotal region for China's agricultural output and an ecologically sensitive zone—have received limited attention. Our investigation reveals that the Songnen Plain, a crucial hub for China's food security as a leading crop-producing area, is increas-

ingly afflicted by prolonged drought episodes. The swift advent of these droughts often leaves scant room for proactive and effective management strategies, leading to significant repercussions for the region's agricultural productivity and socio-economic stability. Despite the inherently high soil moisture characteristic of the Songnen Plain, its tendency toward elevated soil evapotranspiration makes it especially susceptible to arid conditions. The combination of abundant atmospheric moisture with infrequent rainfall during the monsoon season leads to accelerated soil moisture depletion. Simultaneously, the area's dense vegetation further exacerbates this condition by drawing water from deeper soil layers, thus increasing vulnerability to flash drought occurrences.

Our findings highlight an alarming frequency of droughts induced by water scarcity in areas such as Hailun and extensive parts of Qiqihar, coinciding with major rice cultivation zones lacking adequate irrigation infrastructure—a situation that echoes Hunt's observations. Furthermore, heatwave-driven droughts, predominantly recorded along the Songhua River's main course, have occurred more than five times in 60% of the instances between 2002 and 2022. While less common than their water scarcity counterparts, the spatial extent of heatwave-driven droughts cover nearly the entire mainstem area of the Songhua River. Remarkably, the frequency of heatwave-driven droughts has increased in recent years, initially covering the study area extensively at the beginning and potentially evolving into combined heatwave-driven scarcity events. The analysis of inter-annual variability highlights a broader impact scope for water deficit droughts over their heatwave-driven counterparts, likely due to the region's increased agricultural water demands and evapotranspiration rates. Additionally, although there is a general decrease in flash drought occurrences, localized outbreaks have become more rapid.

Vegetation relies primarily on soil moisture for water uptake, which in turn influences transpiration losses, stomatal regulation, and stem water dynamics [48,49]. Drought conditions foster a complex interaction between soil moisture and atmospheric parameters, affecting transpiration and photosynthetic activities and, consequently, the water-carbon flux coupling [37,50]. Beyond mere soil moisture scarcity, ecosystem respiration exhibits increased sensitivity to thermal rises, with the GPP response observed in 95% of the identified flash droughts across the Songnen Plain—highlighting a marked sensitivity to drought conditions. Notably, the GPP response in the Qiqihar segment of the Nenjiang River Basin and the Tao'er River Basin was observed in 91% of drought instances, significantly exceeding the response in the northeastern segment of the Songhua River Basin, where only 65% of droughts triggered a GPP response. This disparity primarily arises from varying vegetation restoration conditions across the plain. The western portion of the Songnen Plain, in particular, showed increased drought susceptibility, suggesting that post-drought soil moisture levels may be insufficient for subsequent vegetation regeneration and development. However, the northeastern basin areas and the Second Songnen River exhibited a lower ecosystem response rate to flash droughts.

The ecological impact of water deficit- and heatwave-induced droughts in the Songnen Plain lasted for 13 and 17 days, respectively. During water deficit-induced droughts, the ecological impact timeframe in over half of the flash drought events within the Ersong watershed and the Nenjiang section was limited to merely 5–8 days—significantly shorter than in the northeastern Songhua River watershed. This extended impact duration suggests the vegetation's limited adaptability to drastic soil moisture changes. The observed increase in WUE and its underlying variant uWUE during drought conditions indicates enhanced ecosystem resilience to these flash climatic shifts. Notably, the Suihua and Qiqihar sections of the Songhua and Nenjiang Rivers, respectively, reported the highest WUE and uWUE values across both drought types, likely due to the superior WUE of forest ecosystems prevalent in these areas. The northeastern regions of the Songnen Plain boasted the highest forest coverage, in contrast to the significantly lower WUE observed in Changchun and Daan, and primarily because of the reduced forest area coverage.

There are some uncertainties in this study. The soil moisture data derived from remotely sensed inversion may not fully account for the complex hydrological interplay

between deep subterranean water reserves and surface moisture levels [51]. Future research should incorporate multi-layer soil moisture observations within an integrated hydrological model framework. Such an approach will provide a more nuanced understanding of soil water balance dynamics, offering deeper insights into the mechanisms driving flash drought evolution and their ecological consequences. In addition, it is evident that if the duration of the flash drought is too brief, it may not be sufficiently long to exert a detrimental effect on the ecosystem. Conversely, a mild flash drought could potentially stimulate vegetation growth [52]. Furthermore, vegetation's response to drought tends to be delayed; the longer the overall drought persists, the more significant its influence on vegetation becomes [53]. Consequently, solely focusing on the rapid development (or burst) phase of a flash drought limits our understanding of its ecological impact [54,55]. It can be anticipated that a subsequent continuous drought following a flash drought could have profound effects on vegetation [56]. However, determining whether this impact should be attributed to a flash or continuous drought and from which perspective to assess the flash drought's impact are questions worth considering.

In addition, it should be noted that the definition of flash droughts and the selection of indicators greatly influence the frequency of flash droughts [13]. Although we used the definition of flash droughts by Yuan et al. [2] found that can characterize the rapid onset and sudden propagation of flash drought events, we probably overestimated or underestimated drought patterns due to a lack of large-scale actual observations for verification. In addition, the selection of SM thresholds can impact the evaluation indicators of flash drought events [34,41,57,58]. The number and duration of flash droughts may vary greatly under different thresholds, despite similar spatial patterns [13]. Furthermore, using different data sources (SM and meteorological variables) to define flash droughts can lead to some uncertainties in deriving their characteristics and potential drivers over a given region [59–61].

## 5. Conclusions

This study, utilizing the NNsm reanalysis dataset, investigated the occurrence, impact, and types of flash droughts in the Songnen Plain from 2002 to 2022, alongside the terrestrial ecosystems' responses. Our findings highlight the urgent need for targeted drought management strategies and further research in this critical area. Below are our main conclusions:

(1) We find an increased frequency and uneven spatial distribution of flash droughts in the Songnen Plain. The increasing frequency of flash droughts with more than seven occurrences was mainly distributed in Baicheng and Songyuan in Jilin and Suihua and Wuchang in Heilongjiang. Flash droughts exhibited a widespread but uneven spatial coverage, with the average duration of 28–30 days.

(2) The region experiences a higher incidence of water scarcity flash droughts and longer durations of heatwave-driven droughts. Both types exhibit a fluctuating upward trend, with water scarcity flash droughts increasing at a faster rate.

(3) The ecosystems show varying responses to these flash droughts in the Songnen Plain. Heatwave-driven flash droughts, predominantly occurring around the Songhua River coast, elicit a slower response rate with weak resilience across most areas. Conversely, ecosystems are more sensitive to heatwave-driven flash droughts, with GPP responding more rapidly compared to water scarcity flash droughts.

(4) There is a noticeable spatial difference in WUE between the development and recovery periods of both flash drought types. Heatwave-driven flash droughts mainly occurred alongside the river, while water scarcity flash droughts are more prevalent in the semi-arid regions of western Jilin.

These findings underscore the complexity of flash drought phenomena and their profound impact on the Songnen Plain's ecosystems. They call for enhanced monitoring and adaptive management strategies to mitigate the adverse effects of such droughts. Future research should focus on the long-term ecological impacts, effective drought prediction



models, and comparative studies with other regions facing similar challenges. Implementing sustainable water management practices and developing resilient agricultural systems are crucial steps toward mitigating the impacts of flash droughts in the Songnen Plain and similar ecosystems worldwide.

**Author Contributions:** Conceptualization, J.S., Y.W. and M.C.; methodology, Q.Z.; software, J.S.; validation, Q.Z. and Q.M.; formal analysis, L.J., J.S. and Q.M.; investigation, L.J. and Q.Z.; resources, Y.W.; data curation, Q.M.; writing—original draft preparation, J.S. and Y.W.; writing—review and editing, M.C. and G.Z.; visualization, C.D., G.Z. and Q.M.; supervision, M.C. and Y.W.; project administration, Y.W.; funding acquisition, Y.W. and G.Z. All authors have read and agreed to the published version of the manuscript.

**Funding:** This work was supported by The National Key Research and Development Program of China (2021YFC3200203), the National Natural Science Foundation of China (421010514, 422070881, U2243230 and 42371169), the Postdoctoral Science Foundation of China (2021M693155 and 2022M723129), and the Strategic Priority Research Program of the Chinese Academy of Sciences and China (XDA28020501 and XDA28100105).

**Data Availability Statement:** The data presented in this study are available upon request from the corresponding author. The data are not publicly available due to privacy.

**Acknowledgments:** We would like to sincerely acknowledge and express our deep appreciation to the editor and to three anonymous referees for their thorough review of this work.

**Conflicts of Interest:** The authors declare no conflicts of interest.

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
