# Peer review of "Spatiotemporal Variation in Water Deficit- and Heatwave-Driven Flash Droughts in Songnen Plain and Its Ecological Impact"

_remotesensing, doi:10.3390/rs16081408_

Round 1
Reviewer 1 Report
Comments and Suggestions for Authors
Review for “Spatiotemporal variation of water-deficit and heatwave driven flash droughts in Songnen Plain and its ecological impact”
The authors analyzed the spatiotemporal variation of two types of flash droughts in Songnen Plain during 2002 to 2022, and then discuss its implication to ecological impact (e.g. GPP, WUE, uWUE) .the text is abundant, and the study is significance to the ecological sensitive regions in China. Some suggestions are for you as follows.
1) Don’t use first person way to depict the abstract text (e.g, “We” in L26, P1). More passive tense sentence should be used
2) What is it means for the conclusion “Additionally, the anomalies in basal water use efficiency …. Within fores-dense regions” in L37~38, P1
3) L120, the time period has some mistake “2020-2022” and what the (26)
4) L133, which refence you referred to “zhang et al”
5) Some descriptions on the heat wave-induce flash drought and water-deficit-induce flash droughts, both of them are described by the high temperature and low soil moisture. What it referred to “evaportransipiration rates surpass typical bounds” and “ evapotranspiration rates falling below the normative spectrum” in L211~216. It seems unclearly, please add the clear and accurate description.
6) L231, the font is wrong for “SIGMAGPP” and “uGPP“
7) L241~257: what is the formula for “WUE”, and what is the difference between “WUE” and “uWUE", why chose two concept rather than Uwue
8) The units should be marked in the figures, eg. Fig2b, fig3b, the bar in Fig4, Fig6.
9) There are no any data information in fig7. What is colorbar
Author Response
Many thanks for your valuable comments and your dedication to the review processes. We appreciate it very much. We have diligently revised the manuscript and provided a detailed point-by-point reply to your comments (highlighted in Yellow). We hope that the revised manuscript will meet the quality of Remote Sensing. Pleased see the attached PDF file.

Reviewer 2 Report
Comments and Suggestions for Authors
This manuscript investigates the spatial patterns of two categories of flash droughts and the response of ecosystems in Songnen Plain. The topic of this article is of interest to the reader. However, there are some points that need to be improved before a possible publication.
My main criticisms come from two aspects: (1) The GPP data only last to 2018 (lines 132-134), how can authors explore the ecological response from 2002 to 2022. In addition, how do authors match the time scale of GPP, 8 days, to other variables, mainly 5 days for flash drought identification; (2) how to identify heatwave-induced and water scarcity-induced flash drought, the authors do not give specific criteria. The authors claim temperature, evaporation, etc., are required for identifying different types of flash droughts, but all flash droughts identified in this manuscript are based on the methods of Yuan et al., which depends only on soil moisture. It comes doubts that whether it is correct to correlate these hydrometeorological variables with the occurrences of flash droughts identified by Yuan et al.
Moreover, there are numerous grammar mistakes in this manuscript.
(1) Line 46: A duplicate The
(2) Lines 73~75: Mo et al., identified the precipitation-deficiency droughts by precipitation, temperature and evapotranspiration (ET), not by soil moisture. The references (13,17) here do not match precipitation-deficiency droughts. Mukherjee et al., (Ref 13 in the manuscript) used root-zone soil moisture, and Christian et al (Ref 17) used ET and potential ET. None of them meets the requirements of precipitation-deficiency droughts.
Refs:
Mo, K.C., Lettenmaier, D.P., 2016. Precipitation Deficit Flash Droughts over the United States. J. Hydrometeorol. 17, 1169-1184.
(3) Line 71: They bifurcated, not them bifurcated
(4) 2 Data and Methodology, the authors should give a figure containing the necessary information of Songnen Plain, e.g., location in China, land uses
(5) Lines 118-120: Surface soil moisture or root-zone moisture, please specify directly which one do you use
(6) Line 120: 2020-2022? What does (26) mean?
(7) Lines 218: The terrestrial GPP
(8) Lines 231-232, using the same mathematic symbols in the equation (3)
(9) Lines 234 to 256, I suggest to use equations to illustrate the calculation of response time and frequency and uWUE, to make these more readable.
(10) Lines 266. Where is the location of Wuchang, I cannot find it in Fig 2.
(11) Figs 2 to 4, give the units of average duration in these figures.
(12) Lines 281, duplicate space
(13) 3.2, still confused, how to identify them from flash droughts identified by Yuan et al.,
(14) Lines 296-298, does this result also comes from Fig.4, I cannot find them. Additionally, please give specific months for different seasons, given that the classification of month in each season may exist some differences for different regions. Please add . at the end of line 298.
(15) Line 302, 2002 to 2021?
(16) Figure 7, no color shown in this figure! What does +- mean in this figure?
(17) Lines 428-430, as shown in Lines 118-120, you have root-zone soil moisture, why you choose to use the surface soil moisture, as Yuan et al., 2019 also used the root-zone soil moisture for flash drought identification
(18) Line 435: tt?
(19) Lines 489-490, duplicate sentence.
Comments on the Quality of English LanguageThere are numerous grammar mistakes that need to be corrected.
Author Response

(The authors gave the same response as above.)

Reviewer 3 Report
Comments and Suggestions for Authors
see attached file.

Comments on the Quality of English Languagesee attached file.
Author Response

(The authors gave the same response as above.)

Round 2
Reviewer 1 Report
Comments and Suggestions for Authors
the authors have addressed all my concerns, and I suggest the manuscript should be published
Author Response
We greatly appreciate your affirmation and the time you have invested. Wishing you all the best.
Jiahao Sun
Reviewer 2 Report
Comments and Suggestions for Authors
I think the paper has improved and my comments have been addressed,but there are some references errors in Lines 79 to 89 needing to correct.
Comments on the Quality of English LanguageThe quality of English language meets the criteria of the Journal, but I suggest authors to check their pdf file before submitting.
Author Response
Many thanks for your valuable comments and your dedication to the review processes. We appreciate it very much. Wishing you all the best.
In addition, we have carefully checked all the contents in the manuscript. Particularly, the references has been double checked. We hope that the revised manuscript will meet the quality of Remote Sensing.
Jiahao Sun